# The Distribution of *Campylobacter jejuni* Virulence Genes in Genomes Worldwide Derived from the NCBI Pathogen Detection Database

**DOI:** 10.3390/genes12101538

**Published:** 2021-09-28

**Authors:** Pedro Panzenhagen, Ana Beatriz Portes, Anamaria M. P. dos Santos, Sheila da Silva Duque, Carlos Adam Conte Junior

**Affiliations:** 1Center for Food Analysis (NAL), Technological Development Support Laboratory (LADETEC), Federal University of Rio de Janeiro (UFRJ), Cidade Universitária, Rio de Janeiro 21941-598, RJ, Brazil; aportes@id.uff.br (A.B.P.); anamariasantos5@yahoo.com.br (A.M.P.d.S.); conte@iq.ufrj.br (C.A.C.J.); 2Laboratory of Advanced Analysis in Biochemistry and Molecular Biology (LAABBM), Department of Biochemistry, Federal University of Rio de Janeiro (UFRJ), Cidade Universitária, Rio de Janeiro 21941-909, RJ, Brazil; 3Collection of Campylobacter, Instituto Oswaldo Cruz, Fundação Oswaldo Cruz, Rio de Janeiro 21040-900, RJ, Brazil; sheila.duque@gmail.com; 4Graduate Program in Food Science (PPGCAL), Institute of Chemistry (IQ), Federal University of Rio de Janeiro (UFRJ), Cidade Universitária, Rio de Janeiro 21941-909, RJ, Brazil; 5Graduate Program in Veterinary Hygiene (PPGHV), Faculty of Veterinary Medicine, Fluminense Federal University (UFF), Vital Brazil Filho, Niterói 24230-340, RJ, Brazil; 6Graduate Program in Sanitary Surveillance (PPGVS), National Institute of Health Quality Control (INCQS), Oswaldo Cruz Foundation (FIOCRUZ), Rio de Janeiro 21040-900, RJ, Brazil; 7Graduate Program in Chemistry (PGQu), Institute of Chemistry (IQ), Federal University of Rio de Janeiro (UFRJ), Cidade Universitária, Rio de Janeiro 21941-909, RJ, Brazil

**Keywords:** *C. jejuni*, WGS, virulence genes, virulence factors, MLST, *flaA*, *flaB*

## Abstract

*Campylobacter jejuni* (*C. jejuni*) is responsible for 80% of human campylobacteriosis and is the leading cause of gastroenteritis globally. The relevant public health risks of *C. jejuni* are caused by particular virulence genes encompassing its virulome. We analyzed 40,371 publicly available genomes of *C. jejuni* deposited in the NCBI Pathogen Detection Database, combining their epidemiologic metadata with an in silico bioinformatics analysis to increase our current comprehension of their virulome from a global perspective. The collection presented a virulome composed of 126 identified virulence factors that were grouped in three clusters representing the accessory, the softcore, and the essential core genes according to their prevalence within the genomes. The multilocus sequence type distribution in the genomes was also investigated. An unexpected low prevalence of the full-length flagellin *flaA* and *flaB* locus of *C. jejuni* genomes was revealed, and an essential core virulence gene repertoire prevalent in more than 99.99% of genomes was identified. Altogether, this is a pioneer study regarding *Campylobacter jejuni* that has compiled a significant amount of data about the Multilocus Sequence Type and virulence factors concerning their global prevalence and distribution over this database.

## 1. Introduction

*Campylobacter* is a Gram-negative bacterium that is able to infect several wild and domesticated animals, contaminated water, raw milk, and animal-based food worldwide [1,2]. Among the species, *Campylobacter jejuni* (*C. jejuni*) is responsible for 80% of human campylobacteriosis and is the leading cause of gastroenteritis globally [3,4]. In addition, it may cause bacteremia, sepsis, meningitis, urinary tract infections, and hemolytic-uremic syndrome. In some cases, the patient may develop complications after infection such as inflammatory bowel diseases (IBD) and Barrett’s esophagus [5]. Post-infectious sequelae are emerging with severe illnesses such as Guillain–Barré syndrome (GBS) [6], a rare autoimmune inflammatory disease of the Peripheral Nervous System, and Miller Fisher syndrome, which is also a rare disease [7].

The relevant public health risks of *C. jejuni* are caused by particular virulence genes encompassing its virulome. Unlike the classical virulence factors such as the T3SS and lipopolysaccharide (LPS) observed in other enteropathogens, *C. jejuni* lacks or does not rely extensively on many classical virulence factors such as enterotoxigenic *Escherichia coli* and *Salmonella* spp. [8,9,10]. It has the particular growth necessities apparent in a fragile organism; however, it has a complex array of fitness and virulence factors [11] that aid the bacterium to respond to the defense mounted by the host (adhesion, invasion, and temporary survival inside the human intestinal epithelial cells (IECs) in vitro). Collective components such as the helical shape, amphitrichous flagella, capsular polysaccharide (CPS), Lipooligosaccharide (LOS), O-linked glycosylation of flagellins, and genotypic and phenotypic variation owing to an augmented number of phase-variable loci are some particularities that promote the successful enteric lifestyle of *C. jejuni* [9]. Some mechanisms of pathogenesis over *C. jejuni*, such as the discovery of the novel autotransporter *CapC* and other virulence factors, have constantly been elucidated [12,13].

*Campylobacter jejuni* is efficient at generating an extensive intraspecies heterogeneity due to its phase variation ability. These properties require sophisticated molecular typing methods to differentiate species and subtypes and elucidate infection sources. Multi Locus Sequence Typing (MLST) has proven to be the molecular method of choice and has provided strong evidence that genetically closely related strains of *C. jejuni* can colonize different hosts [14]. Closely related strains usually belong to large clonal complexes (CC), which share multiple MLST alleles (STs). Moreover, the MLST has provided unified, comprehensive, and portable *Campylobacter* isolate characterization, with curated databases of genotypes available (pubMLST.org/*Campylobacter*) [15]. The genetic diversity of these species is reflected in the more than 11,800 STs registered on the PubMLST database. Notably, more than 8000 *C. jejuni* sequence types were registered by 2020 [16]. Certainly, the determination of *C. jejuni* STs is a valuable resource to the scientific community as a whole for the genotyping investigation of this critical and yet incompletely understood pathogen.

There are web servers and resources that offer free comprehensive analysis pipelines, such as the NCBI Pathogen Detection platform (https://www.ncbi.nlm.nih.gov/pathogens/, accessed on 24 September 2021), the PubMLST (https://pubmlst.org/organisms, accessed on 24 September 2021) and Enterobase (http://enterobase.warwick.ac.uk/, accessed on 24 September 2021) provided that there are no privacy or ethical issues in submitting data to an online repository. The PubMLST seems to be more diverse than the NCBI, with fewer redundant assemblies. In addition, the contributions of USA and UK agencies to genome deposits in the NCBI database are much more significant. The PubMLST also contains the integrated software BIGSdb (https://github.com/kjolley/BIGSdb, accessed on 24 September 2021) that allows a calling engine for a gene-by-gene approach in the same browser, facilitating analysis in real-time. On the other hand, there are already several species-specific databases in the NCBI Pathogen Detection platform against which the isolates of interest can be compared. However, this website does not offer the MLST typing or the virulence genes repertoire for the Campylobacter species.

Studies involving whole-genome sequencing (WGS) have facilitated the biological understanding of gastrointestinal pathogens and several existing microorganisms. The advances of WGS generated by high throughput sequencing platforms has allowed the mass submission of genomes to public repositories in the last years [17,18]. Consequently, in silico studies from bacterial genomes have increased in the last few years, providing several possibilities in gene analysis [19,20,21,22]. The software VirulenceFinder—VFDB (http://cge.cbs.dtu.dk/services/VirulenceFinder/, accessed on 29 June 2021) [23,24] uses WGS data for the in silico characterization of several pathogenic bacteria and enables researchers and clinical health personnel to quickly extract and interpret virulence-relevant information from the genome data [21]. Considering the remaining insights that can still be understood about *C. jejuni* virulence, we designed this study to analyze lots of WGS by an in silico bioinformatics analysis combining epidemiologic metadata to increase our current comprehension of the *C. jejuni* virulome from a global perspective.

## 2. Materials and Methods

### 2.1. Data Collection

We downloaded the entire collection of 61,711 *Campylobacter* genomes assembled until 4th May 2021 from the Pathogen Detection Web Browser at the National Biotechnology Information Center (NCBI) website (https://www.ncbi.nlm.nih.gov/pathogens/, accessed on 4 May 2021). The software KmerFinder v3.2 [25,26,27] was used to identify the *C. jejuni* species within all the *Campylobacter* genomes. The software was installed and run as per the standard instructions informed by the server of the software (https://bitbucket.org/genomicepidemiology/kmerfinder/src/master/, accessed on 20 July 2021).

### 2.2. Multilocus Sequence Type (MLST) Identification

The STs were determined from contigs of each genome using the software Seemann T. mlst v2.19.0 according to the instructions provided by the server of software (https://github.com/tseemann/mlst, accessed on 20 July 2021). The analysis for the *C. jejuni* allele scheme was set from the open-access PubMLST.org website database hosted by the University of Oxford [28].

### 2.3. In Silico Virulome Determination

The bacterial virulome identification was performed according to the instructions provided in the Seemann T. ABRicate software (https://github.com/tseemann/abricate v0.9.9., accessed on 5 August 2021) The virulome was determined by genome hits against the 2597 genes of the virulence database (VFDB) [23,24] using the minimum gene length coverage of 60%. This coverage threshold was set to ensure that genes lying on the edge of a contig or spread over two contigs were not missed due to possible non-perfect assembly [29]. Due to highly variable sequences interspersed among conserved genes and the existence of phase-variable genes such as *motA* and *FlgR* [9,30,31], no thresholds were established for the minimum per base identity to avoid missing any gene.

### 2.4. Statistical Analyses

Statistical analyses were performed using R Studio (http://www.r-project.org, accessed on 12 August 2021) by the Complex Heat map package [32] to cluster and construct abundance heat maps. The prevalence of the virulence genes was calculated by the ratio between each gene and the total number of genes from all the genomes assessed in each geographical region set for this study. The average of the frequencies and the standard deviation were calculated using the formula functions in Microsoft Excel.

## 3. Results

### 3.1. Metadata Overview

Through the KmerFinder software, a taxonomic sequence classifier that assigns taxonomic labels to DNA sequences, we identified and collected all the genomes of *C. jejuni*. Out of the 61,711 *Campylobacter* genomes, a collection of 40,371 *C. jejuni* genomes were finally settled for the virulome investigation. The *Campylobacter* species identified in each genome and the origin of the genomes regarding the geographic location can be seen in Appendix A.

### 3.2. Multilocus Sequence Type (MLST)

The sequence types (STs) were investigated in all the *Campylobacter jejuni* genomes, although a few could not be determined (Appendix A). Over the identified STs, we compiled their global distribution over the five main continents (Appendix A). North America’s genomes presented 745 different STs, followed by Europe (714), Asia (74), South America (64), Oceania (58), and Africa (44). Due to the vast diversity of STs, we ranked the ten most commonly detected STs within this collection of *Campylobacter jejuni* recovered worldwide. Notably, ST50 or ST353 led the STs in all continents except in Asia, where ST2988 was the most common ST identified within the genomes (Table 1 and Appendix A).

### 3.3. Campylobacter Jejuni Virulome

The *Campylobacter jejuni* collection presented a virulome composed of 126 identified genes out of the 2597 virulence genes settled in the virulence finder database—VFDB (Appendix A). Our clustering analysis revealed that the *Campylobacter jejuni* virulomes were grouped in three vertical clusters according to their average prevalence and three horizontal clusters according to their prevalence profiles in each geographic location (Figure 1). The virulence genes from the North America, Oceania, and Europe genomes showed similar prevalence profiles and were grouped in Cluster-C. The virulence genes from the African and South American genomes were grouped in Cluster-B. Finally, the virulence genes from the Asian genomes showed a unique prevalence profile and were grouped in Cluster-A.

Cluster-1 grouped the accessory genes of the virulome with prevalence values ranging from an absence of the genes up to 48.48% (Appendix A). The average prevalence in the cluster was 13.17 ± 2.67%. The virulence factors involved the bacterial capsule biosynthesis and transport, especially by sugar and aminotransferase enzymes (*Cj1421c*, *Cj1422c*, *Cj1426c*, *Cj1432c*, *Cj1434c*, *Cj1435c*, *Cj1436c*, *Cj1437c*, *Cj1438c*, *Cj1440c*, *glf*, *kfiD*), the Type IV secretion system (*virB10*, *virB11*, *virB4*, *virB7*, *virB8*, *virB9*, *virD4*), the immune evasion—LOS (*Cj1137c*, *Cj1138*, *cstIII*), the O-linked flagellar glycosylation (*maf4*), the capsule biosynthesis and transport (*cj1427c*), and the motility and export apparatus (*flaA*, *flaB*). The sugar transferase *Cj1434c* displayed the highest prevalence (25.13%) in the genomes from South America. The African genomes displayed the lowest average prevalence of the cluster (9.21%). The plasmid-mediated genes *virB-D* that encode the Type IV secretion system in *C. jejuni* were absent in the genomes from Africa and Oceania, were very low in prevalence in Europe (0.78%) and Asia (0.60%), and were low in prevalence in North America (3.88%) and South America (4.56%). The gene *cstIII* was also low in prevalence in the genomes from Asia (4.79%). In Oceania, the genomes displayed the lowest prevalence (18.13%) of the sugar-nucleotide epimerase/dehydratase *cj1427c*. The flagellin *flgA*/*flaB* genes displayed the lowest prevalence in Europe (12.47/12.13%).

Cluster-2 grouped the softcore genes of the virulome with prevalence ranging from 62.38 up to 77.44%. The average prevalence within this cluster was 71.08 ± 5.68% (Appendix A). The virulence factors involved the immune evasion—LOS (*gmhA2*, *Cj1135*, *neuA1*, *neuB1*, *neuC1*, *Cj1136*, *wlaN*), the capsule biosynthesis and transport (*Cj1419c*, *Cj1416c*, *Cj1417c*, *Cj1420c*, *hddC*, *hddA*, *fcl*, *rfbC*), and the motility and export apparatus (*ptmA*, *ptmB*, *pseD/maf2*). The gene *neuA1* had a higher prevalence (above 42.11%) in the genomes from North America, Europe, Oceania, South America, and Africa than Asia (23.35%). The genes *Cj1419c*, *Cj1416c*, *Cj1417c*, *Cj1420c* displayed the highest prevalence in all continents (above 83.04%) except in Asia (below 71.86%). The genes *gmhA2*, *hddC*, and *hddA* displayed a lower prevalence in Oceania (below 67.25%) than the other continents (above 77.54%). The genes *neuB1* and *neuC1* displayed a higher prevalence (above 71.04%) in North America than in the other continents (below 64.40%). The genes *wlaN* and *pseD/maf2* displayed the lowest prevalence (46.11/44.14%) in the genomes from Asia and Europe, respectively.

Cluster-3 grouped the core genes of the virulome with prevalence ranging from 71.86 up to 100%. The average prevalence within this cluster was 99.44 ± 0.26% (Appendix A). This cluster grouped the largest repertoire of genes with the smallest standard deviation in their average prevalence. The cluster was mainly formed by the virulence genes responsible for the motility and export apparatus (55 out of 82) as the flagellar basal body rod and hook protein (*flg*), the flagellar biosynthesis and assembly protein (*flh*), the flagellar motor protein (operon *fli*), the chemotaxis protein (*che*), and the flagellar motor protein (*mot*). A group of 32 genes within this virulence factor (*cadF*, *fliL*, *flgG*, *motA*, *flhA*, *pseB*, *kpsE*, *kpsM*, *flgI*, *fliD*, *flgA*, *fliI*, *flgF*, *flgP*, *flaG*, *flgH*, *cheY*, *flgD*, *flgJ*, *flgM*, *flhF*, *flhG*, *hldE*, *fliY*, *fliG*, *fliA*, *ciaC*, *fliF*, *fliH*, *fliS*, *fliE*, *flgB*) encompassed the essential core of the virulome in which the average prevalence was above 99.9%. The flagellar basal body rod protein *FlgB* was detected in all the 40,371 (100%) *C. jejuni* genomes investigated. The genes *fliK*, *pseE/maf5*, *kpsC*, *cdtA*, *pseA*, and *cysC* displayed the lowest average prevalence within the cluster (below 96.52%).

## 4. Discussion

*Campylobacter* is the leading cause of bacterial foodborne gastroenteritis in the world [4]. Within this genus, *C. jejuni* has always been the most relevant clinical [11,33,34], infectious [9], and epidemic zoonotic worldwide [35]. Here we combined more than forty thousand publicly available *Campylobacter jejuni* genomes of strains isolated from more than forty countries of five continents in the world. The synthesized data will help the scientific community advance more in understanding this species pathogenesis. As far as we know, this is a pioneer study regarding *Campylobacter jejuni* that has compiled a significant amount of data about the Multilocus Sequence type and the virulence factors concerning their global prevalence and distribution over this database.

Multilocus sequence typing (MLST) is a gold standard informative tool that has successfully characterized the population structure of *Campylobacter*, identifying lineages such as sequence types (STs) and clonal complexes (CCs) [36,37]. The need for a standard global microbial identification and the power of the MLST explain the many databases and schemes created. The PubMLST (https://pubmlst.org) is the reference database hosting more than 77,000 *C. jejuni* isolates worldwide. Most of the metadata deposited in this database contain only the STs scheme, although the upload of whole-genome sequences has reached more than 40,000 *C. jejuni* isolates. Even though most of the *C. jejuni* ST has already been identified in the PubMLST, our results link them to metadata such as the collection date, the location, the isolation source type of the strains, and the virulence factor composing the *C. jejuni* virulome of the samples derived from NCBI (Appendix A). Moreover, the STs displayed in a word cloud diagram quickly highlight their prevalence and distribution worldwide (Appendix A). Notably, the wide diversity of *C. jejuni* STs in the world is probably due to the plasticity in their genetic relationship once the seven schemed housekeeping genes have shown weak clonality among the *C. jejuni* population [15].

We ranked the top ten identified STs from our genome collection by geographical location (Table 1). The ST50 was the most commonly reported ST in Europe and in Oceania. It was the second most common ST in North America and the sixth most common ST in South America. This sequence type has been recently characterized in *C. jejuni* genomes from North America, Europe, and Australia, highlighting its importance as a global pathogen [16]. The ST50 distribution presented here corroborates with Wallace et al. [16], who showed that the ST50 is among the top ten commonly reported STs within the more than 77,000 sequences deposited in the PubMLST in all continents except Africa. Interestingly, none of the host generalist sequence types ST21, ST45, and ST48 were reported top in Africa, Asia, and South America, although the chicken specialist ST353 [38,39,40] was most frequently identified in Africa and South America. It is also important to comment that the most hyper-aerotolerant, multi-stress resistant, and strong biofilm-producing *C. jejuni* isolates belonged to the host generalist clonal complexes ST21, ST45, ST48, and ST206 [41]. In summary, the frequency and distribution of the STs displayed here only represent the genomes collected from the Pathogen Detection Database for this study. To better understand the prevalence and distribution of the *C. jejuni* STs around the globe, we recommend associating our data with the additional information from the PubMLST database.

A large number of the *C. jejuni* Capsular Polysaccharide (CPS) genes are involved in capsule biosynthesis and transport. Capsule-related genes were first reported at the beginning of the 2000s in the reports of Karlyshev et al. [42] on the surface of *C. jejuni*, found on the outermost layer composed of repeated units of several sugars [43,44]. We identified a group of CPS genes in Cluster-1 (*Cj1421c*, *Cj1422c*, *Cj1426c*, *Cj1432c*, *Cj1434c*, *Cj1435c*, *Cj1436c*, *Cj1437c*, *Cj1438c*, *Cj1440c*, *glf*, *kfiD*) with an average prevalence lower than 15% in the *C. jejuni* genomes worldwide. The remaining CPS genes were grouped in Cluster-2 and 3 and displayed an average prevalence above 70%. The reason for this discrepancy was well understood by the complete genome sequence of the strain NCTC11168 that clarifies the multiple mechanisms of structural variation of CPS, including the exchange of entire clusters by horizontal transfer, gene duplication, deletion, fusion, and contingency gene variation [42]. CPS is also the major sero-determinant of the Penner typing scheme [45], and the high variability of the CPS genes in the *C. jejuni* genome results in a variety of Penner serotypes [46]. Besides the high genetic rearrangements both within and outside the CPS gene cluster, the locus showed a unique mosaic structure maintained in hypervirulent *C. jejuni* clones and is a key virulence factor for the induction of systemic infection and abortion in pregnant animals [47]. Lastly, the mutants Δ*kpsS-C-M* were shown to be deficient in capsule production, resulting in the loss of its ability to induce bacteremia and liver infection in the murine model for the infectivity of *C. jejuni* [47,48]. The high prevalence of the locus *kps*C-T-S-D-F-E-M in our collection of genomes corroborates the importance of these genes for the vitality of *C. jejuni* and provides evidence that they are potential candidates for the development of vaccines against *C. jejuni* worldwide.

The virulence genes *virB-D* are plasmid-mediated, and their prevalence in the *C. jejuni* genomes is associated with the presence of the plasmid *pVir*. The plasmid contains components of a type IV secretion system (T4SS) important for several major bacterial pathogens [49]. However, its involvement in pathogenicity which increases bloody diarrhea in *C. jejuni* enteritis is controversial [50,51,52,53]. The very low prevalence of *vir* genes in humans and broilers found by Iglesias-Torrens et al. [54] also suggests that the *pVir* plasmid is not required for *C. jejuni* to either colonize birds or infect humans. Our results provide a solid demonstration that the plasmid pVir and its virulence vir genes have a very low prevalence (below 5%) in *C. jejuni* globally, corroborating the hypothesis that the T4SS encoded by the *pVir* plasmid may not be relevant for the virulence of *C. jejuni* [52,55].

In 1993, the polymerase chain reaction (PCR) was developed to amplify the *C. jejuni flaA* gene for restriction fragment length polymorphism (RFLP) analysis for genotyping purposes [56]. Later, in 1997, the demonstration of a recombination within and between flagellin loci of natural strains suggests that flagellin gene typing (restriction fragment length polymorphism analysis of PCR-amplified flagellin genes) cannot be considered a stable method for the long-term monitoring of pathogenic *Campylobacter* populations [57]. However, in the same year, a short variable region (SVR) was identified from the sequences of *flaA* and was shown to be conserved among outbreak-related strains [30]. Recent studies have stated that the direct sequencing of the SVR-*flaA* gene is genetically stable over long periods and is helpful for *Campylobacter* typing. It has a similar or higher discriminatory power than MLST, especially in short-term and localized epidemiological investigations [58,59,60,61]. Punctual studies from multiple places worldwide have reported a high prevalence of flagellin A and B in *C. jejuni* [62,63,64,65,66,67,68,69,70,71].

Conversely, our results demonstrate that the flagellin genes *flaA* and *flaB* are the unique genes within the motility and export apparatus virulence factor group that displayed an average prevalence below 35% and a particularly low average prevalence worldwide. These prevalence averages were even lower in genomes from Europe that displayed values below 12.5%. These results corroborate other studies that have investigated the virulence genes of *C. jejuni* from contigs of whole-genome sequences. In a recent study involving 81 *C. jejuni* and *C. coli* strains from Chile, flagellin genes *flaA* and *flaB* had a low prevalence within the genomes (9 out of 81) [72]. In Ireland, 50% of the *C. jejuni* sequenced strains from clinical and poultry sources presented the genes *flaA* and *flaB*. Curiously, in this study, a higher recovery rate of *flaA* and *flaB* previously amplified by a conventional PCR was observed via WGS screening, showing the better resolution of this technique for virulence detection [73]. In a genome-wide association study with *C.jejuni* isolates from Poland, the authors displayed similar results, reporting that the majority of the virulence genes, with rare exceptions such as *flaA*, *flaB*, or *flaE*, were evenly distributed throughout the genomes of all the STs under study [74]. Finally, in a multi-omics approach study that revealed the potential core vaccine targets for *C. jejuni*, *flaA* and *flaB* were not found within the core or essential core virulome of 173 strains [75].

We verified the presence of the *flaA/B* gene in full annotations of 233 complete genome sequences of *C. jejuni* deposited in the RefSeq database of the NCBI (data not shown). This investigation was necessary to rule out the possibility that the low prevalence of the *flaA/B* gene was caused by (1) the gene lying on the edge of contigs, (2) spread over contigs discarded by low quality, or (3) final draft genomes assembled with low completeness/quality. Out of the 233 genomes, the *flaA* and *flaB* genes were annotated in 17 (7.29%) and 4 (1.71%), respectively. We believe some well-known facts can explain these results. First, the intragenomic recombination in the flagellin locus of *C. jejuni* resulted either in the deletion or repositioning (into *flaB*) of sequences initially located in the *flaA* gene [76]. Second, it has been suggested that *flaB* may serve as a gene donor, of which sequences can be introduced through homologous recombination into the *flaA* gene, either to compensate for deleterious mutations or possibly to increase the immunogenic repertoire of a given *C. jejuni* strain [57,76,77]. Third, in the flagellin locus of *C. jejuni*, sequences essential for the transport and assembly of flagellin monomers into flagellin filaments are located in the conserved regions. In contrast, the internal region lacks functional information and is more highly variable and susceptible to homologous recombination [57,78]. The low prevalence of the full-length flagellin *flaA* and *flaB* locus of *C. jejuni* demonstrated here supports the argument against the long-term stability of flagellin gene typing, highlighting the potential low reliability of these methods for investigation into closely related *Campylobacter* strains [79].

It is also worth noting the low prevalence of the gene *wlaN* in Cluster-2. The wlaN gene is responsible for the production of Beta-1,3 galactosyltransferase which is involved in cell wall synthesis, and in the cytotoxin production of *Campylobacter* and the cytolethal distending toxin genes *cdtA*, *cdtB*, and *cdtC* [80]. The product of the *wlaN* gene also shows ganglioside mimicking structures that are putative to be involved in the development of Guillain–Barré syndrome after *C. jejuni* infection [81,82]. The prevalence of the *wlaN* gene is controversial within the literature. A low prevalence was reported in the *Campylobacter* strains from Japan [83], Bangladesh [62], Ireland [84], and Brazil [85], whereas a study in Korea identified the *wlaN* gene among 100% of human and 78.6% of animal *C. jejuni* isolates investigated [86]. The mass screening of *C. jejuni* genomes presented here showed that the average prevalence of this gene in the world is about 56%. Besides, *wlaN* is considered a vital trait gene for *Campylobacter* virulence, and the intermediary prevalence displayed here indicates that their role in the pathogenesis of *C. jejuni* should be further well-evaluated.

Cluster-3 grouped the core genes of the virulome. A group of 32 genes encompassed an essential core with an average prevalence above 99.99% within the 40,371 *C. jejuni* genomes studied here. The intense presence of the strict genes (*cadF*, *fliL*, *flgG*, *motA*, *flhA*, *pseB*, *kpsE*, *kpsM*, *flgI*, *fliD*, *flgA*, *fliI*, *flgF*, *flgP*, *flaG*, *flgH*, *cheY*, *flgD*, *flgJ*, *flgM*, *flhF*, *flhG*, *hldE*, *fliY*, *fliG*, *fliA*, *ciaC*, *fliF*, *fliH*, *fliS*, *fliE*, *flgB*) in this large number of genomes worldwide develops our the understanding of the pathogenic potential of *C. jejuni*. The applicability of the identification of the core genes, for example, is essential for studies where the identification of the virulent bacterial proteins can help the design of potential drugs and vaccines, especially in bacteria such *C. jejuni* that show extensive genetic variations and a smaller core genome than other foodborne pathogens [75]. Another applicability of the identification of core genes is in the fast detection by a PCR-based assay. As we discussed above, PCR assays based on the whole or specific regions of the flagellin genes *flaA* and *flaB* to detect the genotype of *C. jejuni* require a better understanding and confirmation of their functionality. The detection of the gene *flgA* in 99.9% and *flgB* in 100% of the 40,371 *C. jejuni* genomes studied here provide evidence for its possible application on assays such as those discussed above. An insertional mutation *flgA* caused its complete absence in *C. jejuni* NCTC11168 [87]. This explains the presence in all genomes and proves that the gene is essential for flagellar biosynthesis and biofilm formation. Of the 126 genes that encompass the core virulome identified in this study, 59 (~47%) are related to motility, export apparatus, and adherence. They play vital roles in the rotation and switching of the bacterial flagella and are conducive to *Campylobacter* adhesion and biofilm formation [9]. Moreover, it was proven that the flagellar proteins’ export apparatus as a type-III export system in *C. jejuni* is required for the secretion of invasion antigens (*CiaC* virulence protein), essential for host invasion.

## 5. Conclusions

The current study evaluated the prevalence of *C. jejuni* virulence markers in a large number of genomes from a diverse set of locations never investigated in the *Campylobacter* field before. Our data showed the virulence of *C. jejuni* in a new perspective, bringing novel insights to the scientific community. The in silico analysis of WGS revealed a notable diversity in the occurrence of clinically relevant virulence genes, highlighted by the complete or partial lack of the *flaA* and *flaB* loci in a high number of genomes from all the continents. This observation requires further investigation to support the explaining causes we presented above. Altogether, we combined valuable epidemiological and genomic data from several virulence profiles in an extensive collection of *C. jejuni* strains worldwide.

## Figures and Tables

**Figure 1 genes-12-01538-f001:**
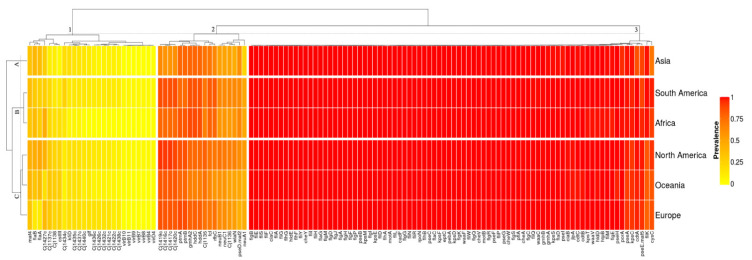
Color heatmap with the 126 virulence genes comprising the *C. jejuni* virulome of the 40,371 whole genome sequences collected from the NCBI Pathogen Detection Database. The prevalence ranges from 0 to 1, in which 0 (yellow) is the least prevalent, and 1 (red) is the most prevalent.

**Table 1 genes-12-01538-t001:** Top ten commonly detected *Campylobacter jejuni* sequence types (STs) recovered from the whole genome sequence in the NCBI Pathogen Detection Database (*n* = 40,371) from each continent.

Rank	North America (*n* = 30,504)		Europe (*n* = 9246)		South America (*n* = 191)		Oceania (*n* = 171)		Asia (*n* = 168)		Africa (*n* = 91)	
	ST	*n* (%)	ST	*n* (%)	ST	*n* (%)	ST	*n* (%)	ST	*n* (%)	ST	*n* (%)
1	353	1602 (5.25)	50	563 (6.09)	353	24 (12.57)	50	29 (16.96)	2988	14 (8.33)	353	9 (9.78)
2	50	1388 (4.55)	5136	416 (4.50)	2993	20 (10.47)	528	13 (7.60)	22	8 (4.76)	362	5 (5.43)
3	48	1260 (4.10)	48	398 (4.30)	8741	15 (7.85)	48	11 (6.43)	985	8 (4.76)	7784	5 (5.43)
4	45	1236 (4.05)	21	383 (4.14)	1359	14 (7.33)	567	7 (4.09)	587	7 (4.17)	52	4 (4.35)
5	982	1220 (3.99)	45	297 (3.21)	475	12 (6.28)	9817	6 (3.51)	2140	6 (3.57)	460	4 (4.35)
6	8	1013 (3.32)	257	273 (2.95)	50	7 (3.66)	2398	6 (3.51)	10,086	5 (2.98)	1038	4 (4.35)
7	806	857 (2.80)	122	239 (2.58)	52	7 (3.66)	658	5 (2.92)	27	4 (2.38)	5	3 (3.26)
8	459	756 (2.47)	19	237 (2.56)	403	6 (3.14)	257	5 (2.92)	986	4 (2.38)	523	3 (3.26)
9	21	697 (2.28)	61	209 (2.26)	463	6 (3.14)	190	5 (2.92)	2209	4 (2.38)	607	3 (3.26)
10	222	674 (2.20)	354	171 (1.85)	883	4 (2.09)	6964	4 (2.34)	5	3 (1.79)	1036	3 (3.26)

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
