# Peer review of "The Distribution of Campylobacter jejuni Virulence Genes in Genomes Worldwide Derived from the NCBI Pathogen Detection Database"

_genes, 2021, doi:10.3390/genes12101538_

Round 1

Reviewer 1 Report

Panzenhagen et al. report in their current manuscript about an in silico analysis of Campylobacter jejuni virulence gene abundance in genomes derived from the NCBI pathogen detection site.

The analysis focuses on country-specific MLST groups and classify virulence gene abundance into three groups (high/medium/low abundance virulence genes). Several of the virulence genes are then discussed. Unfortunately, the analysis doesn't go deeper. In addition, the analysis suffers from the focus on the data from the NCBI pathogen detection site, which is heavily biased towards genomes from the US and the UK (see contributing agencies on https://www.ncbi.nlm.nih.gov/pathogens/about/). Better would be to include genomes from the PubMLST database (also several ten thousands, but more diverse than the NCBI data) as well and sort out redundant assemblies.

The authors try to deliver some country-specific statements, but this can only fail when the number of genomes is as low as n=91 for whole Africa or n=191 from South America! This as well might be better after inclusion of the PubMLST genomes.

The structure of the paper should be streamlined: Parts like the list in L 131 to 148 should be deleted and put into a (supplementary) table. The focus should be on data analysis and presentation of the results, which is way to shallow for a serious analysis right now: Besides including the data from PubMLST, I miss for example a co-occurrence network of virulence genes and maybe a synteny analysis. Also the authors decided to lower the virulence detection gene length coverage to 60% - to include those genes bordering a contig. However, this is problematic since many genes with a low alignment length coverage are adjacent to transposable elements and thus disfunctional, in fact most of the genes at the edge of a contig are disrupted by transposable elements.

I also have a problem how the authors filtered for "pure" C. jejuni genome sequences. I do not understand why they opt for such a strict 100% match (instead of a more loose, let's say, 99.5% match) to C. jejuni, filtering out more than 20.000 genomes. Also the corresponding data should be delivered in a (supplementary) table to get insights which and how many genomes had what % match.

Finally, the title is misleading: One would expect an in depth analysis of flaAB abundance, but the authors just present that there is a low number of these genes in the NCBI database, no insight into why it is that way and if this is related to possibly a lower virulence (maybe these genes were of low abundance in isolates from chicken, but high in rather high abundance from diseased people?). Interestingly, a control using the C. jejuni NCTC11168 FlaA sequence as BLASTp query on the PubMLST database delivered more than 95% positives - however about half of them were below a 60% alignment length coverage, which might be one reason for the low recovery in the current study. This should be further discussed and a genome completeness/quality (e.g. via CheckM and filtering for genomes with a completeness >95%) should be included as well.

The methods are not always clearly described and the used scripts should be delivered as supplement or via a github link and the language/grammar needs a thorough check.

Author Response

Response to Reviewer 1

Point 1. “Unfortunately, the analysis doesn't go deeper”

Authors' response: Yes, we understand the analysis can go much deeper. However, our study compiled too much information with several details that were difficult to focus on in any particular analysis. For example, the found related to the flaA and flab genes were occasionally discovered, and deeper analysis requires a unique study regarding this topic.

Point 2. “In addition, the analysis suffers from the focus on the data from the NCBI pathogen detection site, which is heavily biased towards genomes from the US and the UK (see contributing agencies on https://www.ncbi.nlm.nih.gov/pathogens/about/)”. Better would be to include genomes from the PubMLST database (also several ten thousands, but more diverse than the NCBI data) as well and sort out redundant assemblies”.

Authors' response: That is right. We agree about the heavy number of genomes from the US and UK in the NCBI pathogen detection site. It was not clear that we chose the NCBI because it lacks information over the MLST and the virulence genes of Campylobacter. You can see there is a virulence profile provided for several pathogens but not for Campylobacter. We add a new paragraph in the introduction section to make it more apparent to the readers. Would you please check lines 119 to 131

Point 3. “The authors try to deliver some country-specific statements, but this can only fail when the number of genomes is as low as n=91 for whole Africa or n=191 from South America! This as well might be better after inclusion of the PubMLST genomes”.

Authors' response: Yes, this is true! We appreciate this suggestion. Unfortunately, we were not aware of the availability of such a complete genome repertoire in the PUBMLST database. To the best of our acknowledgment, PubMLST only provided MLST and cgMLST from deposited strains but not WGS. Indeed, we verified and noticed that genomes are much more diverse than the NCBI data and well distributed from places in the world. Also, there are many duplicates of strains deposited in both databases that might be difficult to track to include in new analysis from this study and return this revision in time. We thank you for this valuable information and will keep it in mind for future similar studies.

Point 4. “The structure of the paper should be streamlined: Parts like the list in L 131 to 148 should be deleted and put into a (supplementary) table”.

Authors' response: Thanks for this suggestion. We created a new table summarizing this data and added it to Supplementary table1.

Point 5. “The focus should be on data analysis and presentation of the results, which is way to shallow for a serious analysis right now: Besides including the data from PubMLST, I miss for example a co-occurrence network of virulence genes and maybe a synteny analysis”.

Authors' response: Yes, there is the possibility to create a co-occurrence network of virulence and a synteny analysis, but, as we said above, there are several duplications in both databases that might be fastidious to synteny analysis especially in such a large number of genomes. Moreover, we believe that our results might represent the population of genomes deposited in the NCBI database. We made some changes in the text and title to clarify this.

Point 6.  “Also the authors decided to lower the virulence detection gene length coverage to 60% - to include those genes bordering a contig. However, this is problematic since many genes with a low alignment length coverage are adjacent to transposable elements and thus disfunctional, in fact most of the genes at the edge of a contig are disrupted by transposable elements”.

Authors' response: This is an important observation that we are aware of and was a concern in our analysis. We believe reducing coverage to 60%, as following the reference [29], will also detect genes with low alignment length coverage adjacent to transposable elements. However, we had to choose not to miss any genes regardless they are functional or not. We focused on the gene prevalence and their functions require other investigation.

Point 7. “I also have a problem how the authors filtered for "pure" C. jejuni genome sequences. I do not understand why they opt for such a strict 100% match (instead of a more loose, let's say, 99.5% match) to C. jejuni, filtering out more than 20.000 genomes. Also the corresponding data should be delivered in a (supplementary) table to get insights which and how many genomes had what % match”.

Authors' response: We made a mistake and provided a very confusing sentence. We re-wrote the entire sentence to make it more clear. There was no filtering for pure or strict (100%) matching for C. jejuni detection. All the genomes identified as C. jejuni by the software were selected to study. Please see the corresponding data delivered in supplementary table1 as suggested. Thanks.

Point 8. “Finally, the title is misleading: One would expect an in depth analysis of flaAB abundance, but the authors just present that there is a low number of these genes in the NCBI database, no insight into why it is that way and if this is related to possibly a lower virulence (maybe these genes were of low abundance in isolates from chicken, but high in rather high abundance from diseased people?). Interestingly, a control using the C. jejuni NCTC11168 FlaA sequence as BLASTp query on the PubMLST database delivered more than 95% positives - however about half of them were below a 60% alignment length coverage, which might be one reason for the low recovery in the current study. This should be further discussed and a genome completeness/quality (e.g. via CheckM and filtering for genomes with a completeness >95%) should be included as well”.

Authors' response: We exaggerate pushing this observation to the title. We changed the title to be more reflective of our analysis. The observations of the flaAB abundance require deep analysis that we could not investigate deeper in this study. As noticed, the alignment length coverage is shallow, and we also verified the presence of the flaAB gene within complete genome sequences of C. jejuni in which there is no interference of completeness/quality of the genome. The abundance was also very low in the complete genomes. Also, we provide several hypotheses to explain these results (please see lines 437 -455) and propose in the conclusion that further investigation is required to understand better.

Point 9. “The methods are not always clearly described and the used scripts should be delivered as supplement or via a github link and the language/grammar needs a thorough check”.

Authors' response: We made several modifications thought the entire methods section to clarify our analysis. In addition, we had help from a college that is a native speaker to revise our language/grammar, and we hope it improves this issue.

Reviewer 2 Report

The authors of the present study have analysed over 40,000 C. jejuni genomes to identify the core "virulome" of this bacterial species. While the prevalence of flagellin genes A and B is quite interesting, it is not the primary focus of the manuscript and I suggest that the authors change the title to be more reflective of their entire analysis. 

Figure 1 needs to be edited in some way so that the genes can be more clearly read. Currently, the font is too small. 

The manuscript needs extensive editing for English grammar and correct word usage.  I have provided some examples as evidence but they are not exhaustive.

Line 22  “Campylobacter jejuni (C. jejuni) has been frequently found in broiler meat and is responsible for 80% of human campylobacteriosis, besides being the primary of human gastroenteritis globally.” This sentence has too many different "themes".

Line 25: While the authors have analysed a great deal of sequences, I do not recommend using the word "massive". In my opinion, it is emotive and I suggest the sentence be edited.

Line 43: Use of the word "provoke" is not appropriate. "Cause" or "induce" are recommended. 

Live 45: The use of the word "acquiring" should be changed to "patients may develop bacteremia...."

Line 218:  The use of the word "evidently" is not appropriate. C. jejuni either is or it is not the most common cause of bacteria associated food borne disease.

Author Response

Response to Reviewer 2

Point 1. “The authors of the present study have analysed over 40,000 C. jejuni genomes to identify the core "virulome" of this bacterial species. While the prevalence of flagellin genes A and B is quite interesting, it is not the primary focus of the manuscript and I suggest that the authors change the title to be more reflective of their entire analysis”.

Authors' response: True. We exaggerated pushing this observation to the title. We changed the title to be more reflective of our analysis.

Point 2. “Figure 1 needs to be edited in some way so that the genes can be more clearly read. Currently, the font is too small”. 

Authors' response: Yes. The figure resolution is good, but it should be presented in landscape orientation to be big enough to be read well. We change the figure orientation accordingly. Please check if it is ok. Thanks.

Point 3. “The manuscript needs extensive editing for English grammar and correct word usage.  I have provided some examples as evidence but they are not exhaustive”.

  • Line 22 “Campylobacter jejuni (C. jejuni) has been frequently found in broiler meat and is responsible for 80% of human campylobacteriosis, besides being the primary of human gastroenteritis globally.” This sentence has too many different "themes".
  • Line 25: While the authors have analysed a great deal of sequences, I do not recommend using the word "massive". In my opinion, it is emotive and I suggest the sentence be edited.
  • Line 43: Use of the word "provoke" is not appropriate. "Cause" or "induce" are recommended. 
  • Live 45: The use of the word "acquiring" should be changed to "patients may develop bacteremia...."
  • Line 218:  The use of the word "evidently" is not appropriate. C. jejuni either is or it is not the most common cause of bacteria associated food borne disease”.

Authors' response: Thanks for all corrections provided. We had help from a college that is a native speaker to revise our language/grammar, and we hope this improves the grammar and word usage. Please follow the track changes to see all modifications through the text.